# *Pseudomonas* intra-genus competition determines the protective function of synthetic bacterial communities in *Arabidopsis thaliana*

Anton Amrhein[1], Mingxiao Zhang[2], Stéphane Hacquard[1,3], Anna Heintz-Buschart[2], Kathrin Wippel [1,2]*

1 Department of Plant-Microbe Interactions, Max Planck Institute for Plant Breeding Research, Cologne, Germany, 2 Swammerdam Institute for Life Sciences, University of Amsterdam, Amsterdam, The Netherlands, 3 Cluster of Excellence on Plant Sciences (CEPLAS), Max Planck Institute for Plant Breeding Research, Cologne, Germany

* k.wippel@uva.nl

## Abstract

The plant root microbiota is crucial for nutrient acquisition, development, and disease suppression. Although commensal bacteria display host preference, their beneficial impact on their cognate host and mechanisms of species selection by the plant are still unclear. We use bacterial culture collections derived from the two model species *Arabidopsis thaliana* (*At*) and *Lotus japonicus* (*Lj*) to design synthetic communities (SynComs) and test their protective function upon exposure of *At* Col-0 to the detrimental root-colonizing *At*-derived *Pseudomonas* isolate R401. *Lj*-derived SynComs were fully protective, whereas *At*-derived SynComs displayed full protective activity only towards a R401 mutant impaired in the production of inhibitory exometabolites. The protective phenotypes were associated with a reduced titer of the R401 opportunistic pathogen. In vitro antagonist assays, *in planta* and in vitro bacterial community profiling, as well as strain-swapping and strain-dropout experiments revealed that competition among commensal *Pseudomonas* strains and R401 determines the success of the opportunist, independent of the original host or the phylogeny of the commensals. Furthermore, we determine the carbon utilization potential of these isolates, which may explain the competition with the detrimental strain and the role of host-secreted compounds. Our results provide evidence that intra-genus interactions within SynComs modulate plant health and disease, and that an individual beneficial strain can be sufficient to outcompete an opportunistic relative. This has implications for the successful development of beneficial microbial consortia for agriculture.

## Introduction

Plants establish tight relationships with the microorganisms surrounding them. These diverse microorganisms include viruses, bacteria, archaea, fungi, protists,

which permits unrestricted use, distribution, and reproduction in any medium, provided the original author and source are credited.

**Data availability statement:** Raw reads of amplicon sequencing data are available at the NCBI Sequence Read Archive under BioProject ID PRJNA1162794.

**Funding:** M.Z. is funded by China Scholarship Council (CSC) grant number 202306350027. Funds to S.H. come from a European Research Council consolidator grant (MICROBIOSIS 101089198) and the special priority programme funded by the DFG, https://www.dfg.de): Deconstruction and Reconstruction of the Plant Microbiota (SPP DECRyPT 2125; HA 8169/2-2). K.W. received funding from the German Research Foundation (DFG, https://www.dfg.de) special priority programme Deconstruction and Reconstruction of the Plant Microbiota (DECRyPT SPP2125; WI 4247/2-1) and from the Research Priority Area Systems Biology of the Faculty of Science at University of Amsterdam. Funders did not play any role in the study design, data collection and analysis, decision to publish, or preparation of the manuscript.

**Competing interests:** The authors have declared that no competing interests exist.

**Abbreviations:** ANI, average nucleotide identity; CGCs, catabolic gene clusters; DAPG, 2,4-diacetylphloroglucinol; RA, relative abundance; TSB, tryptic soy broth.

and nematodes, and are collectively referred to as the plant microbiota [1,2]. The structure of the microbiota is crucial for plant health and protection against pathogens [3,4], where an imbalance is associated with susceptibility and disease [5,6]. Plant microbiota diversity and composition is primarily shaped by the environment, i.e., the initial microbial pool in the soil, soil physicochemical properties, geography, and climate [7,8]. The plant host then functions as a filter [9,10], especially belowground, where root-secreted compounds such as carbon sources and signaling molecules act as key determinants driving microbial activity and composition [11]. This so-called rhizosphere effect results in a higher microbial density and lower microbial diversity along the soil-root axis. In addition to these main drivers, host species-specific effects on root-associated bacterial microbiota were shown for plants grown in natural environments [12–14], in greenhouse settings [10,15–19], and in controlled gnotobiotic conditions [17]. Host-specific microbiota composition is likely explained by a signature cocktail of root exudates, a fine-tuned innate immune system, and specific root architecture of individual hosts. Which plant and microbial factors determine these important host-specific communities, and how they are established and maintained remains an active field of research.

The major bacterial taxa in the plant microbiota typically belong to six classes, namely Bacteroidota, Bacillota, Actinomycetota (formerly Bacteroidetes, Firmicutes, Actinobacteria), and alpha-, beta-, and gamma-Proteobacteria [3,9,20]. Notably, both plant beneficial and detrimental bacteria are often found within these groups, and even within the same species [21,22]. Although several microbiota members are detrimental in mono-association experiments under specific laboratory conditions, their detrimental activity is often reduced in a community context, suggesting the existence of mechanisms that suppress disease in nature [23–26]. This can be the result of microbe-microbe interactions, in which antagonism, cooperation, and competition shape community composition [27,28]. In addition, host physiology and environmental changes will affect these dynamics. Nevertheless, ecological frameworks predict that diverse communities are more robust against perturbations, which moreover suggests that beneficial microbial traits may only be exerted in a community context [23,29].

The existence of host species-specific root microbiota and the observation of host preference among commensal bacteria suggest a certain degree of adaptation of the microorganisms to a specific host, i.e., metabolic niche [30]. To what extent the host plant benefits from accommodating adapted, distinct yet taxonomically similar strains, e.g., through biotic stress alleviation, is unclear.

In this study, we addressed if (i) the host origin of isolation of bacterial communities affects protection against a detrimental strain, (ii) how a detrimental strain impacts community composition *in planta* and in vitro, and (iii) if microbe-microbe interactions explain the observed phenomena. We used the previously established bacterial culture collections of isolates from roots of *Arabidopsis thaliana* (*At*) (*At*-RSPHERE, [31]) and *Lotus japonicus* (*Lj*) (*Lj*-SPHERE, [17]) to design taxonomically diverse synthetic communities (SynComs) and test their potential to provide protection against the opportunistic *Pseudomonas brassicacearum* strain Root401 (R401) in *At* Col-0 plants [32,33]. We found that low-complexity SynComs protect against R401 independently

of their hosts of isolation, and that SynCom protective function can be abolished by R401 via exometabolite-mediated inhibition of SynCom members. We also revealed that within-genus competition between *Pseudomonas* commensals and the opportunistic *Pseudomonas* R401 explains SynCom protective success. Therefore, we delineate *Pseudomonas* intra-genus competition as a possible strategy to control *Pseudomonas* pathogen emergence in nature.

## Results

### *At*-derived and *Lj*-derived SynComs differentially suppress R401-induced disease in *Arabidopsis*

*P. brassicacearum* R401 was originally isolated from *At* Col-0 plants growing in agricultural soil and is part of the *At*-RSPHERE culture collection [31]. Upon single inoculation on Col-0 seedlings grown in an agar-based system, R401 causes wilting and disease symptoms ([32]; Fig 1A) that were recently shown to be partially dependent on the production of the lipopeptide brassicapeptin A [33]. To test the potential beneficial effect of bacterial communities on plant performance and resistance, we infected sterile Col-0 seedlings and seedlings co-cultivated with two different 17-member SynComs of *At*-derived strains (AtSC5 and AtSC6) of diverse taxonomic composition (Materials and methods, S1 Table, S1 Fig) with a dilute culture of R401 (S2 Fig). Inoculation of R401 on *At* Col-0 led to a chlorotic and stunted plant phenotype that was mildly alleviated by AtSC5 and AtSC6, indicating that *At*-derived SynComs largely failed to protect plants from R401 (Fig 1A and 1B). We also tested two other SynComs derived from the *Lj*-SPHERE collection (LjSC5 and LjSC6) with similar taxonomic structure to AtSC5 and AtSC6 (S1 Table) and found that both rescued the R401-induced plant growth defect to levels of non-infected control plants (Fig 1A and 1B). The severity of plant wilting and growth impairment was directly linked to R401 colonization capabilities, since bacterial load was generally greater in roots of mock-treated and AtSC-treated plants compared to those treated with LjSC (Fig 1C). These results show that Col-0 can be protected against the detrimental activity of R401 by bacterial communities, and this protection depends on SynCom composition.

### Protective *At*-SynCom function is suppressed by R401 via exometabolite-mediated inhibition of SynCom members

The decreased bacterial load of R401 on Col-0 roots in the presence of other bacteria may be a consequence of microbial antagonism. To test whether individual strains display antagonistic activity towards R401, we performed an in vitro pairwise antagonistic assay in which a producer strain that is dripped on top of a solidified medium produces exometabolites that diffuse into the medium and inhibit the in-agar growth of a target strain, resulting in a clear and quantifiable halo of inhibition ([34]; Materials and methods). We found no profound effect of AtSC5, AtSC6, LjSC5, and LjSC6 strains on in-agar growth of R401, with only one *Bacillaceae* strain (AtRoot131, S3 Fig) and one *Pseudomonadaceae* strain (AtRoot569, details follow further below) displaying inhibitory activity towards R401. However, in the reverse experiment in which R401 was the producer strain, most commensal bacteria (15/17 in AtSC6; 15/17 in AtSC5; 15/17 in LjSC6; 13/17 in LjSC5) were severely inhibited based on diameter measurement of the inhibition zone formed around the R401 colony (Figs 1D and S4). This result corroborates a recent report identifying R401 as the most antagonistic bacterial isolate of the *At*-RSPHERE culture collection [34]. Notably, the same study revealed that >70% of R401 inhibitory activity in vitro was explained by the production of two molecules, namely the antimicrobial 2,4-diacetylphloroglucinol (DAPG) and the iron-chelating molecule pyoverdine that restrict the growth of other bacteria via direct antimicrobial activity and competition for available iron, respectively [34]. When we repeated our inhibition assays using the corresponding R401 Δ*pvdY*Δ*phlD* double mutant impaired in both pyoverdine and DAPG production (R401mut), we confirmed the significant decrease of antagonism toward *At*-derived and *Lj*-derived commensal bacteria used in our study (Fig 1D). Strikingly, although R401mut retained its detrimental activity on Col-0 plants when mono-inoculated onto the seedlings (Fig 1A and 1B), this detrimental phenotype was now fully rescued by both *Lj* and *At* SynComs, indicating that *At* SynCom inability to protect against wild-type R401 (R401wt) reported above (Fig 1A) was causally linked to R401 exometabolite-mediated inhibition of AtSC5 and AtSC6 members (Fig 1A and 1B).

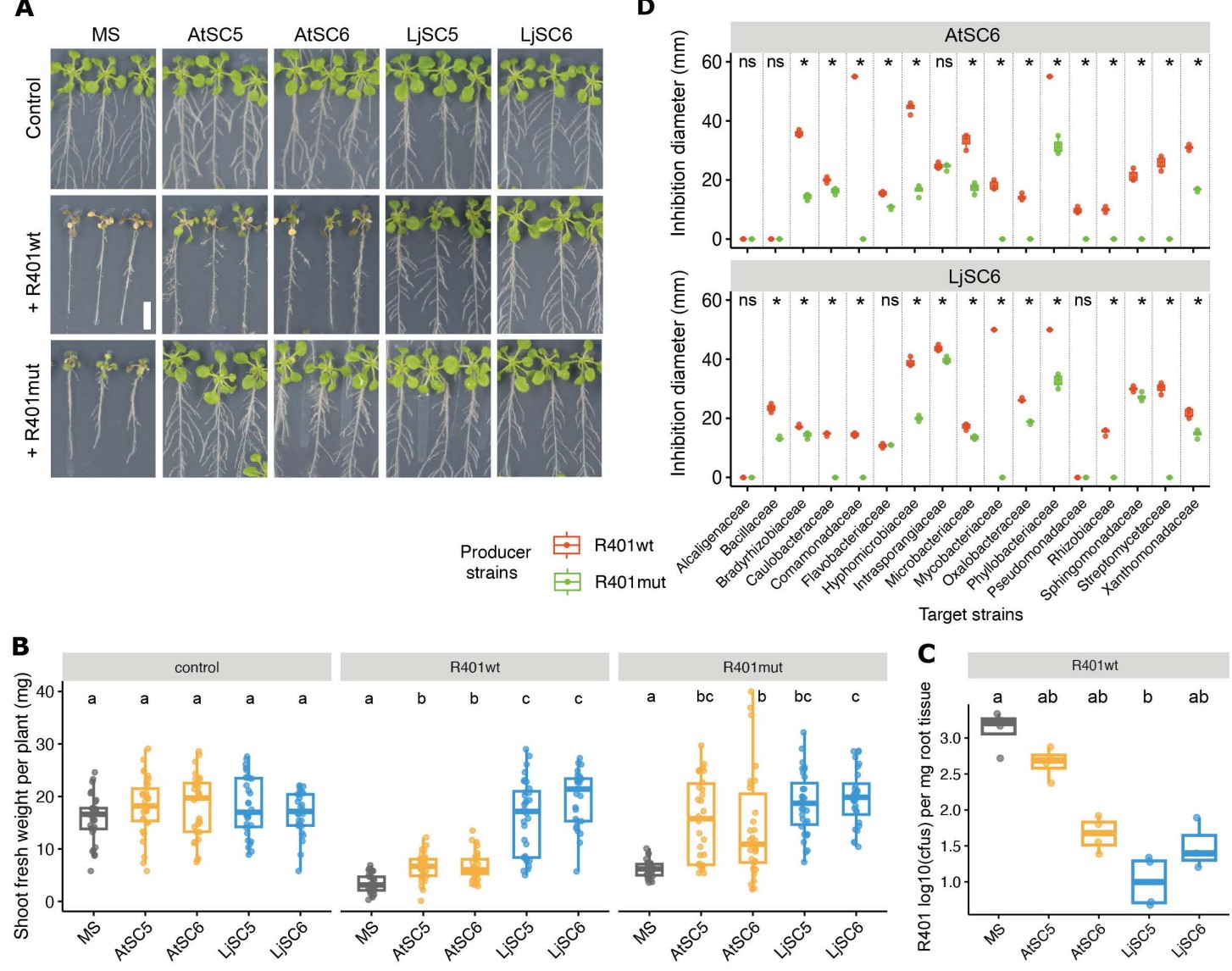

**Fig 1. SynCom-dependent detrimental effect of R401wt and R401mut on Col-0 plants. (A)** Arabidopsis thaliana Col-0 phenotypes grown on MS agar plates alone (MS) or in co-cultivation with the indicated SynComs (AtSC5, AtSC6, LjSC5, LjSC6), and after treatment with mock (Control), R401 wild-type (R401wt) or R401 ΔpvdY ΔphlD (R401mut) strain. Scale bar corresponds to 1 cm. **(B)** Shoot fresh weight of plants shown in A. **e** log10-transformed number of R401wt colony forming units (cfus) at 2 days post-infection on Col-0 roots co-cultivated with indicated SynComs. (B and C) identical letters indicate no significant statistical differences within facetted groups (Kruskal–Wallis followed by Wilcoxon rank sum test and Bonferroni adjustment; *p*-value <0.05). **(D)** Inhibition diameter formed around either a R401wt or a R401mut colony (producers) on a lawn of the indicated individual commensal strains (targets) on an agar plate. Asterisks indicate significant (*p* < 0.05) difference between diameters caused by R401wt and R401mut according to Wilcoxon rank sum test; ns, not significant. The data underlying this figure can be found in S1 Data tabs 1B, 1C, and 1D.

These results show that protection by *At* SynComs depends on the absence of pyoverdine and DAPG production by R401, whereas protection by *Lj* SynComs occurs irrespective of their presence or absence. Based on these findings, we conclude that the differences in the ability to restrict R401 root colonization and detrimental activity depend on SynCom composition.

**R401-induced bacterial community shift is SynCom-dependent and requires DAPG and pyoverdine production**

To assess R401-induced remodeling of SynCom composition *in planta*, we profiled bacterial community composition in Col-0 roots colonized by AtSC6 or LjSC6 one week after infection with either R401wt, R401mut, or a mock solution. 16S rRNA gene amplicon sequencing and subsequent beta-diversity analysis (Bray–Curtis dissimilarities) revealed that R401wt-treated root samples differed from mock-treated and R401mut-treated samples, which cluster together (Fig 2A). The R401wt-induced shift in SynCom composition was clearer for the AtSC6 community compared to LjSC6 (S5A and S5B Fig; 32.8% with p = 0.001 and 12.4% with *p* = 0.12 of variance explained by R401-treatment, respectively, PER-MANOVA), suggesting that LjSC6 is less affected by R401wt invasion than AtSC6. This differential impact of R401 on the communities observed *in planta* could be largely recapitulated in an in vitro experiment with liquid cultures in complex TY medium in which R401 was added either at culture start or to a stabilized SynCom (S6 Fig), revealing that changes in composition were AtSC6-specific (S7A, S7B, S8A, and S8B Figs). This is consistent with the idea that R401-induced bacterial community remodeling is largely host-independent and is therefore primarily explained by direct microbe–microbe competition.

We next calculated the relative abundance (RA) of individual strains within root-associated AtSC6 and LjSC6 and determined those strongly affected by R401. In AtSC6, strains AtRoot83 (*Alcaligenaceae*, log2 FC 2.19), AtRoot935 (*Flavobacteriaceae*, log2 FC 0.45), AtRoot101 (*Intrasporangiaceae*, log2 FC −0.49), AtRoot68 (*Pseudomonadaceae*, log2 FC −0.38), and AtRoot142 (*Rhizobiaceae*, log2 FC 2.85) with RA >1% were significantly affected by R401wt compared to mock treatment (*p* < 0.05) (Fig 2B), and this effect was abolished when R401mut was used. In contrast, none of the LjSC6 strains were significantly affected by either R401wt or R401mut compared to mock-control samples. Of note, the retained or lost in-vitro antagonistic activity of R401mut (Fig 1D) did not correlate with corresponding changes of RA of the commensals *in planta*. In addition, the R401wt-induced community changes of the most abundant strains in AtSC6 *in planta* were similar to the ones in vitro (S8C Fig), but with an additional effect caused by the host environment (S8D Fig). Inspection of R401 levels based on RA in root samples revealed that R401wt efficiently invaded roots pre-colonized by AtSC6, and this colonization phenotype was partially impaired for R401mut (Fig 2C). This indicates that R401 requires production of DAPG and pyoverdine for invading AtSC6 and dominating in roots, consistent with results obtained with a soil-based gnotobiotic system and a different *At*-derived SynCom [34]. However, in contrast to these *in planta* observations, R401wt failed to become a stable member when added to already established liquid in vitro AtSC6 communities (S8A Fig). This may suggest that the host environment constitutes a condition that provides a competitive advantage for R401—possibly related to the spatial structure of the root providing diverse niches—which, however, can be challenged by more competitive commensals. In contrast, both R401wt and R401mut failed to invade LjSC6 pre-colonized roots (Fig 2C).

Taken together, these data indicate that R401-induced bacterial community shift (enabled through production of DAPG and pyoverdine) is required for R401 invasion, and that LjSC6 fully prevents R401 establishment on the roots. This is likely due to the presence of specific strains competing with R401, or possibly through antagonistic activities specifically in the host environment, thereby blocking its root invasion and preventing disease.

**Swapping the *At*-SynCom *Pseudomonas* commensal with the *Lj*-SynCom strain confers protection against R401wt by AtSC6**

We next aimed to narrow down which commensals of AtSC6 and LjSC6 drive R401 root colonization phenotypes and disease outcome. In the AtSC6 context, the RA of the *At* commensal *Pseudomonas* strain AtRoot68 was decreased in the presence of R401wt (Fig 2B), and the opportunist was abundant and caused plant disease symptoms (Figs 1A, 1B, and 2C). However, AtRoot68 remained unaffected within AtSC6 when R401mut was used for infection (Fig 2B). In the LjSC6 context, the RA of the *Lj* commensal *Pseudomonas* strain LjRoot59 was unaffected (*in planta*) or was strongly increased (in vitro), while R401wt levels stayed low and plants healthy (Figs 1A, 1B, 2B, and S8B). Based on these observations,

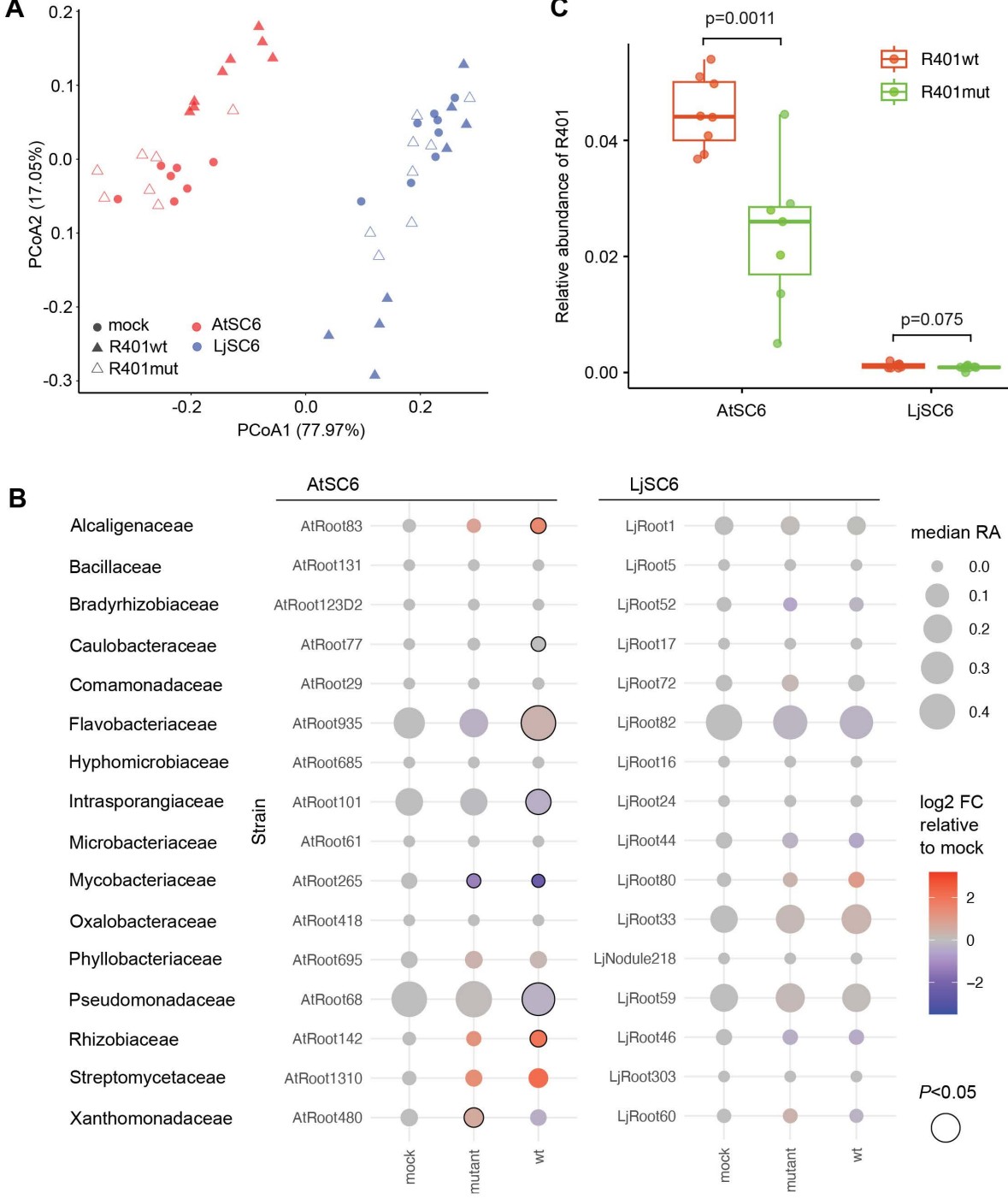

**Fig 2. R401-induced community shifts in AtSC6 and LjSC6 on Col-0 roots. (A–C)** Bacterial community diversity and composition on roots of 21-day old Col-0 seedlings after initial co-cultivation with AtSC6 or LjSC6 and subsequent treatment with either mock, R401wt, or the R401 ΔpvdY ΔphlD mutant at 14 days. (A) Principal Coordinate Analysis (PCoA) of Bray-Curtis dissimilarities of bacterial 16S rRNA amplicon sequences of root samples. (B) Quantification of individual bacterial strains within SynCom AtSC6 or LjSC6. Circle size corresponds to median relative abundance (RA). Color gradient indicates log2-transformed fold change of abundance relative to mock condition (gray indicates no change). Black circle outline indicates significant fold change (Wilcoxon rank sum test, $p < 0.05$). Strain IDs and bacterial families are indicated. (C) Relative abundance of R401wt and R401mut within AtSC6 or LjSC6 on Col-0 roots. $p$-values of a two-sided unpaired Student $t$ test are indicated. The data underlying this figure can be found in S1 Data tabs 2A, 2B, and 2C.

we suspected that competition between individual strains within the *Pseudomonas* genus may determine the colonization success of R401 within a root-associated community. To test this hypothesis, we swapped the two commensal strains in the SynComs and found that LjRoot59 was indeed necessary and sufficient to drive protective function against R401wt *in planta* (Fig 3A and 3B). Normal plant growth was detected after R401wt treatment when LjRoot59 was present in AtSC6 instead of AtRoot68, and plants wilted when AtRoot68 was present in LjSC6 instead of LjRoot59 or after removal of LjRoot59 (Fig 3A and 3B). Amplicon sequencing of root samples and quantification of bacterial RA further corroborated that the presence of LjRoot59 prevents R401wt from establishing as part of the community (S9 Fig). Given that plant performance was significantly better in the presence of LjSC6 even without LjRoot59 compared to axenic plants (Fig 3B), it is possible that other commensal strains within LjSC6 also contribute to protective activity towards R401 *in planta*.

These results show that in taxonomically diverse root-associated communities, whether or not the colonization of an opportunistic pathogen is successful depends on the presence of another individual belonging to the same genus.

## *Pseudomonas*-mediated protection against R401 is strain-specific

Next, we tested 15 *Pseudomonas* members of our two culture collections (S2 Table) to investigate how general or specific the competition between the strains was. In vitro halo of inhibition assays showed that the growth of five strains, irrespective of their host origin, was inhibited by R401wt (including AtRoot68 but not LjRoot59, LjRoot154, and AtRoot569 of the SynComs), but not or partially by R401mut (S10 Fig). Conversely, five other isolates inhibited R401wt growth in vitro (including AtRoot569 but not AtRoot68, LjRoot59, and LjRoot154 of the SynComs), and those five plus four more also inhibited R401mut (Fig 4A). Interestingly, when plants were co-cultivated with these 15 commensal *Pseudomonas* strains separately, 10 of them exhibited protection against R401, allowing the seedlings to reach at least 50% of the rosette area of mock-treated plants (Figs 4B and S11 Fig). This protective ability did not correlate with the host of origin of the strains, nor with their antagonism towards R401 in vitro (Fig 4C). Furthermore, there was no significant correlation between the phylogeny of the strains, which was based on the full-length 16S rRNA gene sequences, and the protection *in planta* (Pearson's product-moment correlation 0.45, $p = 0.092$; S3 Table). We compared the genomic composition of the strains

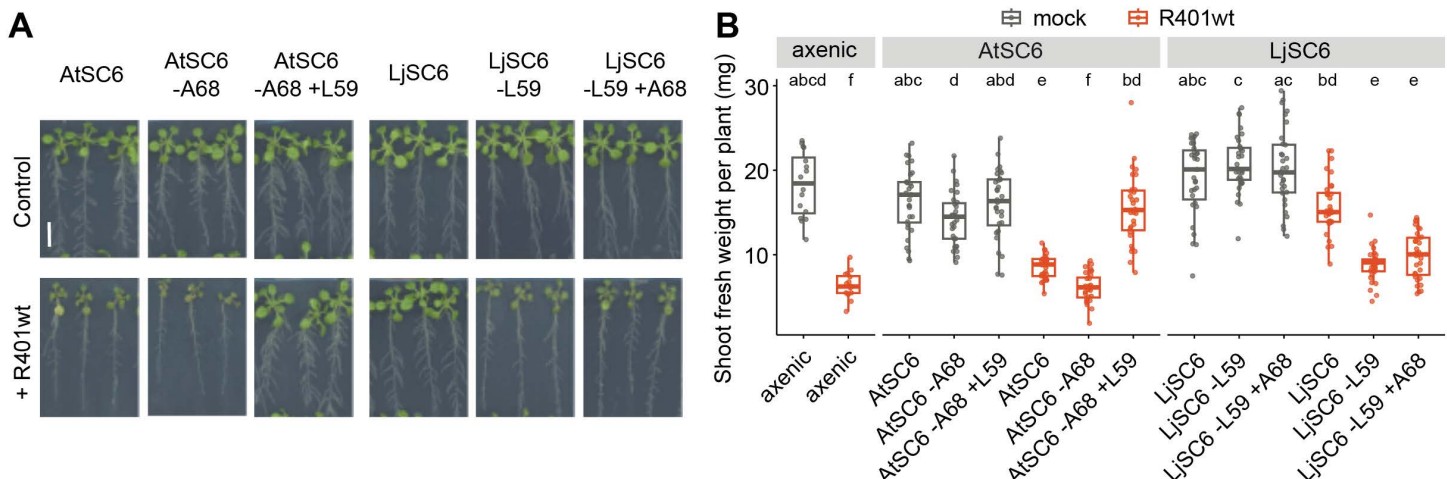

**Fig 3. Rescue of AtSC6 protective activity after *Pseudomonas* commensal swap. (A)** Col-0 phenotypes grown on agar plates in co-cultivation with the indicated SynComs after treatment with mock (Control) or R401 wild-type (R401wt). AtSC6-A68, AtSC6 without AtRoot68; AtSC6-A68+L59, AtSC6 without AtRoot68 but with LjRoot59; LjSC6-L59, LjSC6 without LjRoot59; LjSC6-L59+A68, LjSC6 without LjRoot59 but with AtRoot68. Scale bar corresponds to 1 cm. **(B)** Shoot fresh weight of plants shown in C. Identical letters indicate no significant statistical differences (Kruskal–Wallis followed by Wilcoxon rank sum test and Bonferroni adjustment across all categories; p-value <0.05). The data underlying this figure can be found in S1 Data tab 3B.

[https://doi.org/10.1371/journal.pbio.3002882.g003](https://doi.org/10.1371/journal.pbio.3002882.g003)

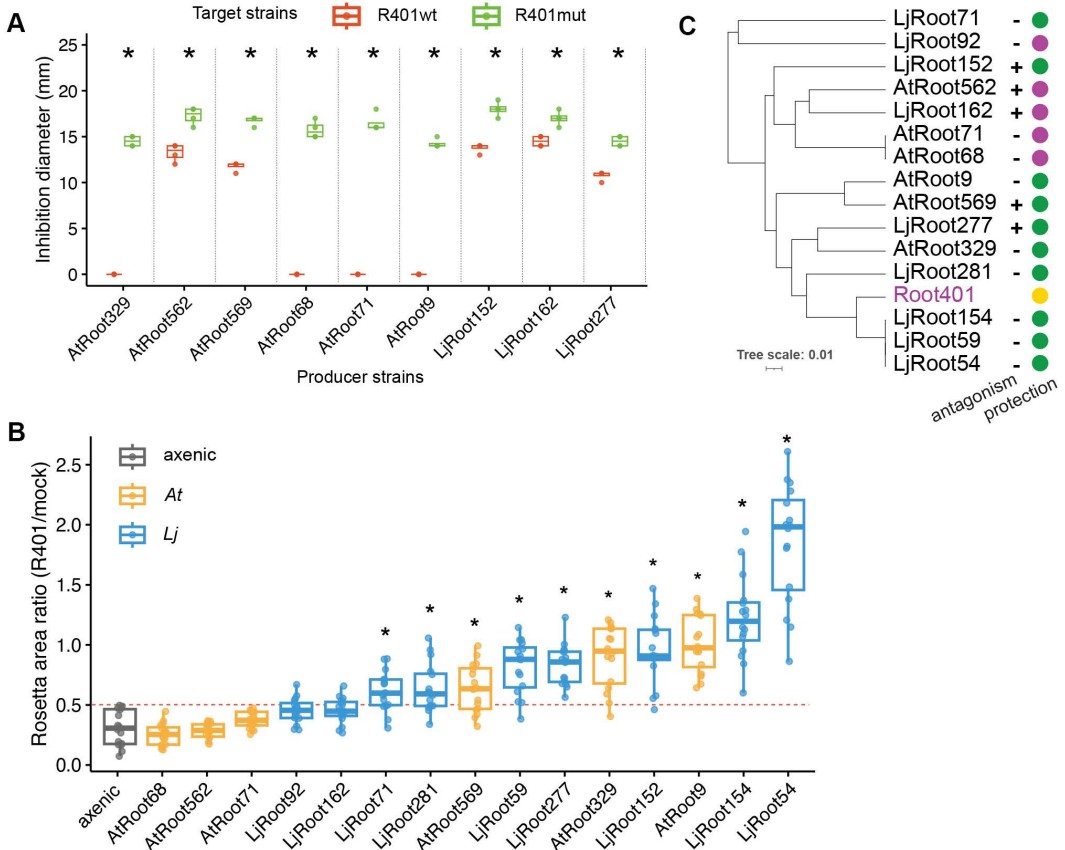

**Fig 4. Competition between R401 and individual *Pseudomonas* strains in vitro and *in planta*. (A)** Inhibition diameter formed around colonies of the indicated nine individual *Pseudomonas* commensal strains (producers) on agar plates on a lawn of either R401wt or R401mut (targets). The other six *Pseudomonas* commensals did not inhibit R401 growth. Asterisks indicate significant ($p < 0.05$) difference between diameters caused by R401wt and R401mut according to Wilcoxon rank sum test. **(B)** Col-0 plants had been co-cultivated with indicated *Pseudomonas* isolates for two weeks, then treated either with R401 or a mock solution. Rosette area of R401-treated relative to mock-treated plants at 21 days. Asterisks indicate significant difference to axenic condition according to Mann–Whitney U test ($p < 0.05$). Dotted red line indicates the 0.5 threshold, where the rosette area of R401-treated plants is 50% of that of mock-treated plants. *At*-derived strains, blue box plots; *Lj*-derived strains, yellow box plots. **(C)** Phylogenetic tree of *Pseudomonas* strains based on full-length 16S rRNA gene sequence. Presence (+) or absence (−) of antagonistic activity by the commensal strains towards R401wt in inhibition assays is indicated. Filled circles indicate protection (green) and no protection (magenta) of Col-0 by these strains against R401wt when in binary *in planta* assays (panel B). The data underlying this figure can be found in S1 Data tabs 4A and 4B, and in S1 File.

more globally by calculating the average nucleotide identity (ANI) between them. We found that protective function is not explained by general ANI similarity to R401 (Wilcoxon rank sum test, $p > 0.85$; S12 Fig; S4 Table), indicating that specific genomic features may explain the *in planta* phenotype.

Of note, two of the commensals phylogenetically close to R401, LjRoot54 and LjRoot154, were slightly detrimental on Col-0 based on the shoot appearance, however, not to the same extent as R401, and the disease phenotype did not become stronger upon R401 infection, indicating that they were still able to outcompete R401.

### *Pseudomonas*-mediated protection against R401 is predicted to occur via resource competition

The variation in the observed competitiveness of the *Pseudomonas* strains *in planta* could be a result of strain-specific variation of growth advantage and proliferation. Given that protective ability did not correlate with the strains' sensitivity or

resistance to R401*,* we hypothesized that resource competition might explain *Pseudomonas*-mediated protection against R401.

We first measured the growth of these strains in a medium supplemented with carbon sources abundantly secreted into the rhizosphere by plants (artificial root exudates: glucose, fructose, sucrose, citric acid, succinate, lactate, glutamate, alanine, and serine; [17,35]). From the growth curves (S13A and S13B Fig), we derived the maximum growth rates and found that they vary among the *Pseudomonas* isolates and do not correlate with protective activity against R401 *in planta* (Wilcoxon rank sum test, $p = 0.76$) (S13C Fig, S5 Table).

Since the *Pseudomonas* strains can generally use these synthetic exudates, their differential competitiveness on roots may be due to their capacity to use other compounds. Therefore, we used rhizoSMASH (https://git.wur.nl/rhizosmash) to predict the bacteria's catabolic potential based on their whole-genome sequences. In total, we found catabolic gene clusters (CGCs) for the catabolism of 34 distinct compounds (S14 Fig, S6 Table). The number of these CGCs varied between 0 and 4 per compound across the *Pseudomonas* genomes (S14 Fig). We measured how similar or dissimilar the CGC profiles of the strains were compared to that of R401 (S15 Fig) and tested if a higher similarity correlated with protective activity *in planta*. We found a moderate correlation (Pearson's product-moment correlation 0.563, $p = 0.029$) between the protective activity and the similarity of the genomic potential for catabolic activity between the protective strains and R401 (S7 Table). In addition, the distances of CGC profiles of protective strains to R401 were significantly smaller than the distances of non-protective strains (Wilcoxon rank sum test, $p = 0.027$).

These results may suggest that bacteria with a similar catabolic potential as R401 may possess a competitive advantage in co-cultivation *in planta*, thereby representing one possible mechanism by which *Pseudomonas* commensals might restrict R401 colonization on the roots.

## Discussion

Within the concept of the holobiont [36], the microbiota is considered a genetic and thus functional extension of the host gene repertoire. Therefore, it is reasonable to assume that plants benefit from accommodating a tailored subpopulation of micro-organisms. The microbial species may in turn experience an advantage from colonizing a plant species that provides a specific structural and chemical environment. In our study, we investigated if taxonomically diverse SynComs of different host origin were able to provide protection against the detrimental activity of the opportunistic *Pseudomonas* strain R401 on roots of *At* Col-0.

We found that within a controlled agar-based growth system, the presence of *Lj*-derived strains prevented the plants from wilting after infection with R401, whereas taxonomically equivalent *At*-derived strains failed to do this efficiently (Fig 1A and 1B). The disease phenotype correlated with a high detectable titer of R401 on the roots (Figs 1C and 2C), which could be a result of antagonistic activity of the strain towards the commensals, rendering it more competitive and dominant. R401 was originally isolated from healthy-looking Col-0 plants as part of the *At* culture collection [31]. Its dominance is possibly more pronounced within the *At* SynCom since it may have developed a competitive advantage among other *At*-associated bacteria. Consistent with that, R401wt-induced remodeling of AtSC6 composition was directly linked to both invasion and disease and requires production of pyoverdine and DAPG (Figs 1 and 2). Additionally, in a previous study and a similar experimental setup, R401wt root colonization was also only slightly restricted by commensal *At* SynComs [32], although these SynComs consisted of only four members and did not contain *Pseudomonas* strains. However, since R401wt inhibited in vitro growth of both *At*- and *Lj*-derived diverse bacterial taxa (Fig 1D), in vitro antagonism is not sufficient to predict dominance *in planta*.

Relative quantification of the bacteria within root-associated SynComs revealed the impact of R401 on the other community members (Fig 2B). A significant effect of R401 on a diverse 18-member SynCom has also been reported for a soil-based system [34]. In those experiments, however, no other *Pseudomonas* strain was included, and R401 did not cause wilting symptoms despite high RA. Interestingly, in our system, the abundance of the commensal *At Pseudomonas*

strain AtRoot68 was strongly decreased in AtSC6, and the *Lj Pseudomonas* counterpart LjRoot59 stayed unaffected, while R401wt failed to colonize within LjSC6 (Fig 2C). Since here the *Pseudomonadaceae* members make up at least 30% of the abundance within the communities, we regarded this differential impact of R401wt as important. We suspected that intra-genus dynamics could play a decisive role for R401wt colonization success. Indeed, when testing the protective function of several individual *At* and *Lj Pseudomonas* isolates *in planta* (Fig 4), we found that other *Pseudomonas* commensals were also capable of suppressing the detrimental effect of R401wt on *Arabidopsis*. However, this trait was independent of the strains' origins, phylogeny, and in vitro antagonism towards R401wt (Fig 4C). These results suggested that, besides the activity of secreted antimicrobial compounds, intra-genus competition may determine colonization success.

Besides direct antagonisms, competition for resources contributes to microbial population dynamics [37] when the amount of nutrients is limited, or when only certain types of nutrients are available. Organisms with a corresponding resource-usage profile then have a competitive advantage over those with more inapt profiles, and the latter could benefit from cross-feeding on metabolic products secreted by other species. In line with this, it was demonstrated for phyllosphere-associated microbiota members that bacteria-bacteria interaction outcomes *in planta* can be predicted by the carbon source utilization profiles combined with genome-based metabolic modeling of the strains and the overlap of those [38]. This study found mostly negative interactions between strains in vitro, which corresponded to niche overlap. An increase of positive interactions *in planta* (and other aspects not covered by the model) was explained by cross-feeding, additional metabolic preferences, signaling, host immune response, and fluctuating environmental conditions. Moreover, Pacheco-Moreno and colleagues recently inferred from the different gene sets of *Pseudomonas* strains isolated from the rhizosphere of two different barley cultivars distinct bacterial preferences for primary plant metabolites, and linked those to the abundance of certain metabolites in the root exudates of the plant genotypes [30]. When we inspected the genomes of our *Pseudomonas* isolates, we found that strains with protective activity against R401 *in planta* possess similar CGCs as R401 itself (S14 Fig, S6 and S7 Tables), which may be a first prediction for a possible niche overlap also in our microbiota reconstitution system. In future experiments, the presence of the corresponding CGC-related compounds in root exudates and their role in competitive root colonization should be explored.

Host involvement in interactions between plants and diverse *Pseudomonas* strains has also been described for a phyllosphere community [22]. In our study, R401wt and commensal dynamics are affected by the *At* host as well, since R401wt invasion of the root-associated AtSC6 was possible, whereas invasion within an established liquid community failed (S8 Fig). The relevant plant factors remain to be shown, and likely comprise root-exuded compounds, including nutrients and signaling molecules that affect both the commensals and the detrimental strain, and parts of the plant immune system.

Our findings corroborate results from other studies on protective activities of commensal strains against pathogenic or opportunistic bacteria in roots, and this is often linked to microbe-microbe dynamics. For example, several studies with tomato and the bacterial wilt-causing strain *Ralstonia solanacearum* demonstrated that microbial competition can lead to the successful exclusion of this pathogen, e.g., correlating with the number [39] and diversity [40] of co-inoculated rhizobacterial strains and the number of utilized carbon sources. Using resource competition networks for SynComs of non-pathogenic *Ralstonia* spp., Wei and colleagues showed that, when all species can use resources similarly, and the resource niche overlaps with the pathogen's, pathogen titer and thus disease decrease [41]. Similarly, in the human gut, commensal bacterial strains only protected against pathogenic invaders if they, in combination with others, exhibit a community nutrient utilization profile that is competitive with the pathogen's [42]. However, if a single strain exhibits such a profile, it may be sufficient for this nutrient blocking. Importantly, our strain-swapping and -dropout experiments showed that LjRoot59 is sufficient to protect Col-0 from being colonized by R401wt (Fig 3). Of note, we cannot exclude that a combination of mechanisms, including resource competition, induction of host immune response, or *in planta* expression of antimicrobials may cause the restriction of R401wt root colonization. Nevertheless, our data indicates that even within a diverse community of different bacterial taxa, one specific strain with relevant properties can outcompete a detrimental one.

 

In addition to R401, other plant opportunistic *Pseudomonas* strains have previously been isolated from healthy plants and other environments [43,44], supporting the idea that the microbiota members exert both antagonistic and competitive activities to keep the pathogen titers low in natural environments. For the detrimental strain N2C3, it was shown that, among diverse *Pseudomonas* isolates, only highly related *Pseudomonas* strains protect *At*, and that they require a functional response regulator ColR that allows for root colonization [45]. In contrast to R401 (Figs 1C and 2C), N2C3 keeps its colonization ability even in the presence of a protective strain [45], indicating different modes of competition of these two detrimental strains, although the different experimental setups may play a role as well. Interestingly, an avirulent mutant of the pathogen, in which the genes for the phytotoxic agent were deleted, was not able to protect plants against the wild-type strain, suggesting the requirement of additional genetic components, despite the already high phylogenetic similarity [45]. Furthermore, besides the intra-genus competition reported by us and by others, individual distantly related bacterial microbiota isolates have been shown to protect plants against pathogens, albeit through mechanisms other than direct resource competition [46]. In line with this, other taxa than *Pseudomonadaceae* may contribute to protective activity in our system as well, since LjSC6 leads to slightly better plant performance than axenically grown plants, even without the *Pseudomonas* commensal LjRoot59 (Fig 3B).

In summary, our experiments show that an individual beneficial strain can be sufficient to outcompete an opportunistic relative, either alone or within a taxonomically diverse microbial community. This is important because it may indicate that, in field settings under certain circumstances, (i) the beneficial species can persist, and (ii) the surrounding members can stabilize the community as a whole. Understanding such dynamics is crucial for engineering stable and beneficial microbial consortia [47]. The difference in RA of certain commensals and opportunistic strains in a competitive situation is likely due to selective carbon source preferences and niche overlap. This concept provides important implications for sustainable agricultural processes. In further studies, especially in greenhouse or field conditions, testing the function of additional host-specific micro-organisms in protection of their host against other, non-adapted but detrimental microbial species will be necessary to determine generality of our findings.

## Materials and methods

### Bacteria

All commensal bacteria were obtained from the *At*-RSPHERE and *Lj*-SPHERE culture collections [17,31]. The compositions of the SynComs are listed in the S1 Table. In brief, 17 strains were chosen per SynCom, representing the 17 bacterial families shared between the two culture collections. The tested *Pseudomonas* strains are listed in the S2 Table. Transgenic Root401 and Root401 *pvdY phlD* mutant both carrying a chromosomal gentamycin resistance, were kindly provided by Felix Getzke [34]. Bacteria were grown at 25 °C in 15g/L tryptic soy broth (TSB) or TY (5g/L tryptone, 3g/L yeast extract, 10 mM $CaCl_2$), with 15g/L Bacto Agar (Difco) for solid media. Gentamycin was added where required to a final concentration of 50 μg/mL.

### Co-cultivation of plants and bacteria on agar plates

*Arabidopsis thaliana* Col-0 wild-type seeds were surface-sterilized by incubation in 70% ethanol for 5 min, followed by two steps of 1-min incubation in 100% ethanol and five steps of washing in sterile MilliQ water. Seeds were kept in approx. 200 μL of water to imbibe and in the cold and dark for stratification for four days. Bacterial liquid cultures were prepared by inoculating 3 mL of 50% TSB medium or TY medium with a colony of bacteria recently streaked onto agar plates. After 2 days of growth, cells were harvested by centrifugation, washed twice in 10 mM magnesium sulfate ($MgSO_4$), and resuspended in 3 mL of $MgSO_4$. OD600 was measured, and SynComs were prepared such that each strain in the mix had a final OD600 of 1. Aliquots of 300 μL were taken of each SynCom, and of 50 μL of each individual strain, frozen and kept at −80 °C until further processing for sequencing. To prepare agar plates, SynComs were added to autoclaved, hand-warm 0.5× MS medium (2.22 g/L MS including vitamins (Duchefa), 0.5 g/L MES, pH adjusted to 5.7 with KOH) containing

10 g/L Bacto agar (Difco Agar, Becton Dickinson) to a final OD600 of 0.005. Per 12 cm × 12 cm squared petri dish, 50 mL of medium only or medium-SynCom mix were used. Twenty sterilized seeds per plate were placed onto the solidified medium in two rows, plates sealed with M3 Micropore tape and placed vertically in a light cabinet with a 10/14 h light-dark cycle. After two weeks, plates were opened inside a clean bench, 15 mL of mock (10 mM $MgSO_4$) or R401 suspension (OD 0.0001 in 10 mM $MgSO_4$) were carefully added per plate, and plates tilted to distribute the liquid. After 10 min of incubation, the liquid was removed and seedlings transferred to new, sterile 0.5× MS agar plates. Samples for viable counts of R401 were taken at 2 dpi (days post-infection), and samples for amplicon sequencing at 7 dpi. Pictures of plants were taken around 7 and around 14 dpi. See S2 Fig for schematic workflow.

## Viable colony counts of R401

Plant shoots were cut off the roots with a scalpel on the agar. Roots of 2–4 plants were transferred to one well of a 12-well plate with sterile 10 mM $MgSO_4$ and washed for a few seconds by swirling them with forceps. Roots were dried by padding on a piece of sterile Whatman paper, transferred to a prepared sterile 2-mL screw-cap tube containing grinding beads, weighed, and put aside on a rack. Four replicates per plate were harvested, and this was repeated for all plates. Roots were homogenized by subjecting the tubes to 2× 6,200 rpm with a 15-s break in the Precellys Homogenizer. 400 µL of 10 mM $MgSO_4$ was added to each tube, and the homogenization procedure repeated. In a 96-well PCR plate, 100 µL of each root extract were transferred to one well, and serial dilutions of $10^{-1}$, $10^{-2}$, $10^{-3}$, $10^{-4}$, and $10^{-5}$ were prepared in 10 mM $MgSO_4$. Ten µL per condition and dilution were spotted onto a 50% TSB squared plate containing 50 µg/mL gentamicin, and the plate tilted such that the liquid was distributed across half the width of the plate. Two technical replicates were prepared, and colony-forming units (cfus) of transgenic, gentamicin-resistant R401 counted after 2–3 days. Only two of the SynCom strains, Flavobacteria AtRoot935 and LjRoot82, are somewhat resistant to gentamycin, meaning that they grow on the selection plates. However, the bright orange color of those colonies could be distinguished from the white R401 colonies.

## Sampling and amplicon sequencing

Plant shoots were cut with a sterile scalpel and forceps and weighed one by one. Roots of 3–4 plants were removed with sterile forceps and washed briefly by swirling in one well of a 12-well plate containing sterile 10 mM $MgSO_4$. Roots were padded on sterile Whatman paper to dry and transferred into Lysing Matrix E tubes (FastDNA spin kit for soil, MPBiomedicals), and frozen in liquid nitrogen immediately. With four samples per plate, and two replicate plates per condition, there were eight replicates per condition in total. Samples were stored at −80 °C. DNA extraction, library preparation, and amplicon sequencing of the 16S rRNA v5-v7 gene region with the in-house Illumina MiSeq platform at the MPIPZ Department of Plant–Microbe Interactions was performed as described previously [17]. Multiplexing of samples was performed by double-indexing with barcoded forward and reverse oligonucleotides for 16S rRNA v5-v7 sequence amplification.

## Analysis of sequencing data

Amplicon sequencing data analysis was performed as described previously [17]. Raw data was demultiplexed according to their barcode sequence using the QIIME pipeline [48]. Merged paired-end reads were quality-filtered and aligned to reference 16S rRNA sequences extracted from the whole-genome assemblies of each individual strain included in a SynCom, using Rbec (v1.0.0) [49]. From the generated count table, RAs were calculated and employed for downstream analyses in R (v4.0.3) with the R package *vegan* (v2.5–6). Data was visualized with the *ggplot2* package (v3.3.0) [50]. For the *in planta* experiment, there were 10.1 mio filtered reads with a median of 16.7k reads per sample; for the in-vitro culturing experiment, there were 741.5k filtered reads with a median of 12.8k per sample.

In silico depletion of R401 was achieved by removing sequencing reads for R401 from the count table prior to computation of RAs. This approach was applied to amplicon sequencing data from samples with plants treated with R401 or

a mock solution for the Principal Coordinate Analysis of Bray-Curtis dissimilarities and for the abundance plot and fold change calculation.

## In vitro inhibition assays

The potential of a producer strain to inhibit the growth of a target strain through the production of diffusible antimicrobial compounds was tested. This was done by spotting the producer strain on top of solidified agar in which the target strain was growing and assessing the formation of a halo (lack of growth of the target strain) around the producer strain. Cultures of individual bacterial strains were grown for two days in TY medium. Strains were harvested by centrifugation, washed twice in 10 mM $MgSO_4$, and adjusted to OD600 of 0.5 in TY medium. To prepare agar plates with the target strains, 50 μL of bacterial culture were added to 50 mL of autoclaved, hand-warm TY agar to obtain OD600 of 0.0005, and the mix was distributed into four round petri dishes (6-cm diameter). After solidification, 10 μL of the producer strain with OD600 of 0.1 were dropped in the middle of the plate and left to dry. Plates were incubated at 25 °C for 2–5 days, and halo formation was recorded by taking pictures of the plates with a scanner. Halo size ("inhibition diameter") was measured as the diameter of the halo crossing the spotted producer strain. Fiji software was used when possible, and manual measurement with a ruler was applied when growth of the target strain was weak and contrast adjustments in the software were not sufficient to visualize the halo on a digital image.

## In vitro co-cultivation of SynComs with R401

To assess the effect of R401 on the composition of bacterial SynComs after growth in culture, SynComs were incubated with or without R401 in liquid medium for multiple days. See S6 Fig for schematic workflow. Bacterial strains were grown in liquid TY medium for 3 days, then diluted 1:10 in fresh medium and grown for another day. Strains were harvested by centrifugation, washed in 5 mL 10 mM $MgSO_4$, and resuspended in 2–5 mL $MgSO_4$, based on visual assessment of culture density. All 17 strains of a specific SynCom were combined in a 50-mL Falcon tube, using 1 mL per strain. OD600 was measured, SynComs diluted to OD600 of 0.1 in a 50-mL tube, and split into two cultures. To one tube, R401 was added to a final OD600 of 0.006, which corresponded roughly to the OD600 of each individual strain in the mixture. Aliquots of 300 μL were taken of all conditions (AtSC6, AtSC6+R401, LjSC6, LjSC6+R401) for time point $t$0, which were profiled as input samples in the amplicon sequencing. Each culture was divided into 4× 5-mL cultures in culture tubes and incubated shaking at 25 °C. After three days, 300 μL each were taken by pipetting from all cultures and frozen at −80 °C. Then, OD600 was measured of all cultures and cultures diluted in new tubes with TY to a final OD600 of 0.1. For the cultures without R401, two batches were prepared, and the amount of freshly grown R401 culture added to one batch for a final OD600 of 0.006. Cultures were incubated shaking at 25 °C for another three days. Aliquots of 300 μL of all tubes were taken and frozen at −80 °C. DNA of all cultures was isolated using alkaline lysis. For this, samples were left to thaw at room temperature. Wells of a 96-well PCR-plate were filled with 20 μL buffer 1 (25 mM NaOH, 0.2 mM EDTA, pH 12), 12 μL of sample were added, the plate incubated at 95 °C for 30 min, 20 μL of buffer 2 (40 mM Tris-HCl at pH 7.5) added to each well, and the plate sealed and kept at −20 °C until further processing (library preparation and amplicon sequencing, see above).

## Phylogenetic trees

Multiple sequence alignment of full-length 16S rRNA gene sequences of Lj and At SynCom strains was performed using the MUSCLE method, and maximum likelihood phylogenetic trees were constructed using the Mega 11 software (https://www.megasoftware.net/). The number of bootstrap replicates was 500. Multiple sequence alignment of the full-length 16S rRNA gene sequences of the *Pseudomonas* strains listed in the S2 Table was performed using the MUSCLE method in the CLC Workbench software. Hierarchical clustering was done with the UPGMA method. Phylogenetic trees were visualized with the help of the Interactive Tree of Life webtool [51].

## Average nucleotide identity analysis

ANI between *Pseudomonas* genomes was analyzed using the FastANI method [52].

## Bacterial growth curves

To measure growth curves, a minimal medium containing M9 salts (3 g/L $KH_2PO_4$, 0.5 g/L NaCl, 1 g/L $NH_4Cl$), vitamins (0.4 mg/L 4-aminobenzoic acid, 1 mg/L nicotinic acid, 0.5 mg/L calcium-D-pantothenate, 1.5 mg/L pyridoxine HCl, 1 mg/L thiamine HCl, 0.1 mg/L biotin, 0.1 mg/L folic acid, 0.4 mg/L riboflavin, 0.2 mg/L cyanocobalamin, 0.05 mg/L lipoic acid), 1 mM $MgSO_4$, 0.1 mM $CaCl_2$, and carbon sources was used. The 5× stock of M9 salts was adjusted to pH 7.2 with NaOH before usage. The 100× vitamin mix was adjusted to pH 6.8 with KOH before usage. First, individual carbon sources were prepared such that the final concentration in the medium was 30 mM with respect to number of C atoms in the molecule. Stock solutions of the carbon compounds were prepared as 10× solutions. For the artificial root exudates, nine carbon sources (glucose, fructose, sucrose, citric acid, succinate, lactate, glutamate, alanine, and serine) were added in equal volumes to the medium, corresponding in total to 1/10 of the volume of the final medium. Bacterial cultures were grown overnight in 50% TSB medium. Cultures were then washed once in minimal medium without carbon sources and OD600 adjusted to 0.5 in minimal medium. In a microtiter plate, 20 μL of each strain were added to a well containing 180 μL of minimal medium with carbon sources, yielding an OD600 of 0.05. Plates were incubated shaking at 25 °C. Absorbance at 595 nm was measured with a plate reader (TECAN Infinite F50) at different intervals. The maximum growth rates for each individual strain were calculated using the *fit_easylinear* function of the R *growthrates* package (https://github.com/tpetzoldt/growthrates).

## Detection of catabolic gene clusters in bacterial genomes

Whole genome assemblies based on Illumina short read sequencing for the bacterial isolates are available from previous studies [17,31]. We subjected these assemblies to the rhizoSMASH algorithm (available at https://git.wur.nl/rhizosmash, Yuze Li and Marnix Medema, Wageningen University and Research, the Netherlands) to detect CGCs based on sequence similarity to publicly available bacterial genomes. CGC names with significant hits were extracted from the output data for each *Pseudomonas* strain. Euclidean distances were calculated based on the presence and number of the CGCs within the genomes relative to the R401 genome using the *dist* function of the R core *stats* package.

## Supporting information

**S1 Fig. Phylogenetic trees of SynComs (A) SC5 and (B) SC6.** Tree is based on full-length 16S rRNA gene sequences of *Lj* and *At* SynCom strains (see Materials and Methods). Bootstrap number was 500. The SynComs were designed to exhibit high taxonomic diversity, and to represent the culturable root-associated community of *A. thaliana* and *L. japonicus* (i.e., one representative strain for each bacterial family shared between the *At*-RSPHERE and *Lj*-SPHERE collection; Wippel and colleagues 2021). The data underlying this figure can be found in S2 and S3 Files.
(PDF)

**S2 Fig. Schematic showing the workflow of the infection assays.** See Methods section for details. Created in BioRender. Wippel, K. (2025) https://BioRender.com/i0y55bb.
(PDF)

**S3 Fig. Antagonistic potential of SynCom members towards R401.** In vitro inhibition assay with AtRoot131 (upper row) and AtRoot68 (lower row) as producer. Left column, growth on TY medium only; middle column, growth on a bacterial lawn of R401wt, with slightly visible inhibition halo formed around the AtRoot131 colony; right column, growth on a bacterial lawn of R401mut, with well visible inhibition halo.
(PDF)

**S4 Fig. In vitro growth inhibition of SC5 strains by R401.** Inhibition diameter formed around either a R401wt or a R401mut colony (producers) on a lawn of the indicated individual commensal strains (targets) on an agar plate. Asterisks indicate significant ($p < 0.05$) difference between diameters caused by R401wt and R401mut according to Wilcoxon rank sum test. The data underlying this figure can be found in S1 Data tab S4.
(PDF)

**S5 Fig. Community diversity on Col-0 roots.** Beta-diversity of (**A**) *At* and (**B**) *Lj* SynComs after community establishment on Col-0 roots in the agar-based growth setup. Constrained Principal Coordinate Analyses (CPCoA; constrained by treatment) are shown for Bray-Curtis dissimilarities between 16S rRNA gene amplicons amplified from root samples of three-week old Col-0 plants that had been co-cultivated with the SynComs for two weeks, then infected with mock, R401wt, or R401mut, and grown for another week. Variance explained by infection is 32.8% (PERMANOVA, $p = 0.001$) and 12.4% ($p = 0.12$) for AtSC6- and LjSC6-co-cultivated plants, respectively. The data underlying this figure can be found in S1 Data tabs S5A and S5B.
(PDF)

**S6 Fig. Schematic showing the workflow for the co-cultivation in liquid cultures.** Cultivation of commensal strains with or without R401wt, and time points of sampling. See Methods sections for details. Created in BioRender. Wippel, K. (2025) https://BioRender.com/i0y55bb.
(PDF)

**S7 Fig. R401wt-induced community shifts within in vitro cultures of AtSC6 and LjSC6.** (**A and B**), Principal Coordinate Analysis (PCoA) of Bray-Curtis dissimilarities of 16S rRNA gene amplicons of liquid culture samples from SynComs (**A**) AtSC6 and (**B**) LjSC6. SynComs were grown in liquid medium for three days with or without R401wt (green and blue symbols, respectively). Then R401wt was added or not to the initially non-treated samples (orange and blue symbols, respectively). Samples were taken and profiled after three (t1) and after six days (t2). See S6 Fig for detailed experimental setup. The data underlying this figure can be found in S1 Data tabs S7A and S7B.
(PDF)

**S8 Fig. Community composition in liquid cultures.** Relative abundance (RA) of individual commensal bacterial strains within (**A**) AtSC6 and (**B**) LjSC6 after growth for three (t1) and six (t2) days in liquid culture with or without R401wt added from the start or after three days, based on 16S rRNA amplicon reads. The composition of the communities at the start of the experiment is shown as t0. **C,** Quantification of individual bacterial strains within SynCom AtSC6 after six days (t2). Circle size corresponds to median relative abundance (RA). Color gradient indicates log2-transformed fold change of abundance relative to mock condition, where R401 was not added at t0 (gray indicates no change). Black circle outline indicates significant fold change (Wilcoxon rank sum test, p < 0.05). Strain IDs are indicated. **D,** Principal Coordinate Analysis (PCoA) of Bray-Curtis dissimilarities of 16S rRNA gene amplicons from AtSC6 of liquid culture samples (magenta symbols) or root-associated samples (green symbols) from the *in planta* experiment (Fig 2B), in presence of R401. Shapes depict samples from liquid cultures after three days (t1, triangles) and after six days (t2, squares). The data underlying this figure can be found in S1 Data tabs S8A, S8B, S8C, and S8D.
(PDF)

**S9 Fig. Relative abundance of R401wt in dropout SynComs.** Relative abundance based on 16S rRNA gene amplicon reads of R401wt within the indicated SynComs established on roots of three-week old Col-0 plants that had been co-cultivated with the SynComs for two weeks, then infected with R401wt, and grown for another week. AtSC6_A68do (do, "dropout"), AtSC6 without AtRoot68; AtSC6_A68do_L59, AtSC6 without AtRoot68 but with LjRoot59; LjSC6_L59do, LjSC6 without LjRoot59; LjSC6_L59do_A68, LjSC6 without LjRoot59 but with AtRoot68. The data underlying this figure can be found in S1 Data tab S9.
(PDF)

**S10 Fig. In vitro growth inhibition of individual *Pseudomonas* isolates by R401wt or R401mut.** The inhibition diameter of the halo inhibition assay with either R401wt or R401mut as producers and the *Pseudomonas* commensals as targets is shown. The other 10 isolates were not inhibited. The data underlying this figure can be found in S1 Data tab S10. (TIF)

**S11 Fig. Plant phenotype after co-cultivation with individual *Pseudomonas* isolates and R401 infection. A,** Col-0 phenotypes in co-cultivation with the indicated individual *Pseudomonas* strains after treatment with mock or R401wt. Scale bar corresponds to 1 cm. Images follow the order of the strain IDs in Fig 4B. **B,** Correlation between mean ratio values of rosette area of R401-treated relative to mock-treated plants from experiments performed in Cologne (exp1) and in Amsterdam (exp2); Pearson's correlation coefficient r and p-value are shown. The data underlying this figure can be found in S1 Data tab S11B. (PDF)

**S12 Fig. Average nucleotide identity (ANI) between *Pseudomonas* isolates.** ANI is shown as percentage similarity between each pair of strains. Filled circles indicate protection (green) and no protection (magenta) of Col-0 by these strains against R401wt when in binary in planta assays (Fig 4). (PDF)

**S13 Fig. Growth of *Pseudomonas* strains in ARE. (A and B)** Growth curves of indicated *Pseudomonas* isolates in minimal medium supplemented with ARE (artificial root exudates: glucose, fructose, sucrose, citric acid, succinate, lactate, glutamate, alanine, and serine). **(A)** Strains that protect plants against R401wt. **(B)** Strains that fail to protect plants against R401wt. **(C)** Corresponding growth rates. Color code next to the strain ID indicates protective activity against R401wt *in planta.* Filled circles indicate protection (green) and no protection (magenta) of Col-0 by these strains against R401wt when in binary *in planta* assays. The data underlying this figure can be found in S1 Data tabs S13AB and S13C. (PDF)

**S14 Fig. Bacterial catabolic potential.** Number of catabolic gene clusters (CGCs) related to the catabolism of the indicated compounds in individual *Pseudomonas* strains, based on a survey of their whole-genome sequences using rhizoSMASH (see Materials and methods). The dendrogram shows the hierarchical clustering of the strains based on a distance matrix of the number of CGCs within each strain. Filled circles indicate protection (green) and no protection (magenta) of Col-0 by these strains against R401wt when in binary *in planta* assays (see Fig 4). The data underlying this figure can be found in S6 Table. (PDF)

**S15 Fig. Euclidean distances between R401 and the other *Pseudomonas* strains.** Based on the type and number of catabolic gene clusters (CGCs) detected in the bacterial whole-genome sequences. Distances are shown as numbers within the plot. The data underlying this figure can be found in S7 Tables. (PDF)

**S1 Table. SynCom compositions.** (PDF)

**S2 Table. *Pseudomonas* isolates.** (PDF)

**S3 Table. Distance indices of *Pseudomonas* isolates to R401 16S gene.** (PDF)

**S4 Table. Average nucleotide identity (ANI) between *Pseudomonas* genomes.** (PDF)

**S5 Table. Growth parameters of individual *Pseudomonas* strains.**
(PDF)

**S6 Table. Catabolic gene clusters detected in *Pseudomonas* genomes.**
(PDF)

**S7 Table. Euclidean distances of R401wt to the other *Pseudomonas* strains based on the presence of catabolic gene clusters.**
(PDF)

**S1 Data. Data Tables.**
(XLSX)

**S1 File. Data File for Fig 4C.**
(TXT)

**S2 File. Data File for S1A Fig.**
(TXT)

**S3 File. Data File for S1B Fig.**
(TXT)

## Acknowledgments

We would like to thank Dieter Becker and Michel Schulz for valuable technical help. We thank Yuze Li and Marnix Medema (Wageningen University and Research, the Netherlands) for making rhizoSMASH available. Thanks to José Flores-Uribe for continued availability for computational questions, and to Paul Schulze-Lefert for constructive feedback.

## Author contributions

**Conceptualization:** Kathrin Wippel.

**Formal analysis:** Anton Amrhein, Mingxiao Zhang, Anna Heintz-Buschart.

**Funding acquisition:** Kathrin Wippel.

**Investigation:** Kathrin Wippel.

**Methodology:** Anton Amrhein, Mingxiao Zhang, Anna Heintz-Buschart, Kathrin Wippel.

**Resources:** Stéphane Hacquard.

**Supervision:** Kathrin Wippel.

**Validation:** Kathrin Wippel.

**Visualization:** Anton Amrhein, Mingxiao Zhang, Kathrin Wippel.

**Writing – original draft:** Kathrin Wippel.

**Writing – review & editing:** Anton Amrhein, Mingxiao Zhang, Stéphane Hacquard, Anna Heintz-Buschart, Kathrin Wippel.

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
