## [Editor Report · Decision Letter 0]

Dear Kathrin,

Thank you for submitting your manuscript entitled "Pseudomonas intra-genus competition determines protective function of SynComs in Arabidopsis thaliana" for consideration as a Research Article by PLOS Biology. I was happy to see this one in.

I have now discussed the manuscript with an academic editor with relevant expertise and I am writing to let you know that we would like to send your submission out for external peer review. However, we think that the study should be considered as a Short Report.

Before we can send your manuscript to reviewers, we need you to complete your submission by providing the metadata that is required for full assessment. To this end, please login to Editorial Manager where you will find the paper in the 'Submissions Needing Revisions' folder on your homepage. Please click 'Revise Submission' from the Action Links and complete all additional questions in the submission questionnaire. IMPORTANT. Please, when adding the rest of the metadata choose "Short Report".

Once your full submission is complete, your paper will undergo a series of checks in preparation for peer review. After your manuscript has passed the checks it will be sent out for review. To provide the metadata for your submission, please Login to Editorial Manager (https://www.editorialmanager.com/pbiology) within two working days, i.e. by Oct 04 2024 11:59PM.

Best wishes,

Melissa

Melissa Vazquez Hernandez, Ph.D.

Associate Editor

PLOS Biology

---

## [Decision Letter · Decision Letter 1]

Dear Kathrin,

Thank you for your patience while your manuscript "Pseudomonas intra-genus competition determines protective function of SynComs in Arabidopsis thaliana" was peer-reviewed at PLOS Biology. It has now been evaluated by the PLOS Biology editors, an Academic Editor with relevant expertise, and by several independent reviewers.

In light of the reviews, which you will find at the end of this email, we would like to invite you to revise the work to thoroughly address the reviewers' reports. The reviewers are overall positive but have raised several issues that need to be addressed before we can consider the study further. R1 questions specifically the statement that plant performance is improved over axenic growth with the LjSC6 SynCom and requests either more evidence, or for it to be removed. R2 wonders about the genetic content and the presence of pvdY and phlD genes in some of the Pseudomonas strains. R3 want some clarifications and an additional statistical analysis in the in vitro assays shown in Fig 1D. R4 would like to see if effect of R401 WT on AtSC6 composition are similar in plant roots as in liquid culture. All reviewers would like to see some modifications in the text and the figures. We agree with these concerns and require additional experimental revisions and textual changes to address them, as these improvements will strengthen the work.

Given the extent of revision needed, we cannot make a decision about publication until we have seen the revised manuscript and your response to the reviewers' comments. Your revised manuscript is likely to be sent for further evaluation by all or a subset of the reviewers.

**IMPORTANT - SUBMITTING YOUR REVISION**

*Re-submission Checklist*

*Published Peer Review*

*PLOS Data Policy*

*Blot and Gel Data Policy*

Sincerely,

Melissa

Melissa Vazquez Hernandez, Ph.D.

Associate Editor

PLOS Biology

REVIEWERS' COMMENTS:

Reviewer #1:

In this study the authors examine the biocontrol potential of two synthetic communities (syncoms) derived from the model species Arabidopsis thaliana (At) and Lotus japonicus (Lj) against the root pathogenic Pseudomonas isolate R401. They show that the Lj-derived SynComs were able to protect Arabidopsis, whereas the At-derived SynComs could not, unless antimicrobial operons in R401 were deleted. They then conducted a series of in planta and in vitro competition and biocontrol assays, alongside bacterial community profiling, strain-swap and strain-dropout experiments, and determined that competition between R401 and commensal Pseudomonas strains, independent of the original host or the phylogeny of the commensals was responsible for biocontrol. They suggest that similarities in carbon utilisation potential may explain the efficacy of competition for some commensal isolates with the detrimental strain.

This is an interesting and elegant study that advances our understanding of the relationship between pathogenic and commensal/biocontrol microbes in the rhizosphere microbiome. I have a few comments and queries, outlined below:

1. The statement on line 70-72 would benefit from a reference, e.g. Pacheco-Moreno et al. PLOS Biology 2024. (ref 39).

2. Line 141 and Fig 1D: The authors could note here that the LjSC5/6 Pseudomonadaceae were not inhibited by R401, in contrast to the AtSP5/6 populations, which is likely connected to the suppressive activity of this community.

3. Figure 2B is quite difficult to understand. Why is it that major changes in strain abundance are not accompanied by similar changes in circle size? Why are some circles, e.g. for Streptomycetaceae from AtSC6, quite red in colour but are not considered significantly different from the mock? Also, why are some circles dark grey and others light grey? Could the authors perhaps add some more context in the figure legend?

4. Line 220: bacterial RNA?

5. Line 222: The data in Figure 3B do not support the statement that plant performance is significantly improved over axenic growth by the LjSC6 SynCom (the datasets appear to have been produced and analysed separately, so cannot be directly compared in this way). This statement should either be supported with additional data that shows a significant difference between these plants in a contiguous experiment, or removed.

6. Line 225: I don't know how the authors draw this conclusion. Their data does not support this and indeed argues against it - the presence of a specific, suppressive pseudomonad in LjSC5/6 prevents R401 colonisation success, rather than promoting it.

7. Figure 4: The panels in Fig 4B do not appear to be in either numerical or phylogeny order, which makes interpreting this figures rather difficult. Also, the lower panel is unlabelled. Using light green to signify both mild biocontrol and also mild pathogenicity in Fig 4C is confusing. A fourth colour (light purple?) could be used for the mild pathogens for clarity.

8. The legend of Figure 5 is not informative and should be expanded.

9. Figure 5: The logical extension of this analysis is that upon deleting the two biocontrol operons from R401, AtRoot68 is able to compete for nutrients with R401mut and acts like LjRoot59. How similar is the metabolic profile of AtRoot68 from R401?

10. Line 320: DAPG

11. Line 436, 447, 508 and elsewhere, ug/uL is used in place of microgram/microlitre. Replace with the appropriate symbol in each case.

12. Line 487-497: The amplicon sequencing and subsequent data analysis could be explained in a little more detail (e.g. sequencing depth, analysis programs used), rather than simply referenced elsewhere.

Reviewer #2:

The article by Amrhein et al. explores the disease phenotypes on Arabidopsis colonized by the opportunistic pathogen strain Pseudomonas R401 in a variety of synthetic communities. These synthetic communities are comprised of taxonomically diverse strains from Lotus japonicus and Arabidopsis thaliana. Results show that Lotus-derived but not the Arabidopsis-derived SynComs effectively protect Arabidopsis plants from R401-induced wilting. Through a variety of community profiling and growth assays it was determined that this is likely due to competitive dynamics among Pseudomonas strains. Strain Root 68 from the Arabidopsis SynCom is not effective at protecting Arabidopsis, but Strain Root 59 from the Lotus SynCom is. Additional Pseudomonas strains are tested and using catabolic gene clusters predicted from genomes of these strains the authors conclude that resource competition and niche overlap by strains with similar CGCs to R401 better support pathogen exclusion/resistance. The data underscore the potential for individual beneficial strains, like LjRoot59, to outcompete pathogens in rhizosphere communities, providing insights into resource competition and plant-microbe dynamics that may drive healthy microbiome community structure in real-world plant microbiomes.

Major Comments:

I think it would be helpful to describe briefly how/why the SynComs were constructed around lines 99-104. For example it is not clear immediately that these are 17 member SynComs (and not 5 and 6 member SynComs from the SynCom5 and SynCom6 naming). It may be appropriate to include a phylogenetic tree of the four SynComs somewhere in a figure (main or supplementary).

I take issue with the description of the protection as "SynCom-specific" (line 110). Also line 150 "based on these finding we conclude that there are SynCom-specific differences….". Also line 216 "SynCom protective function". Also at lines 310-312 where protection is cast as a property of the Lj SynCom but not the At SynCom. The authors show that protection is primarily related to the Pseudomonas strains in the SynComs (most clearly shown in Figure 3), so I don't think that this protection should be cast as some property of the SynCom as a whole. Maybe saying that the protection phenotype is only observed in particular communities would be more appropriate or being more explicit that the composition of the SynCom matters for the protection ability.

Are the strains that are protective phylogenetically closer to R401? Is the Average Nucleotide Identity of their genomes closer? The genetic content of the particular Pseudomonas strains that are protective/not are not explicitly mentioned. It is partially covered in passing with the CGCs prediction analysis which suggests that strains that have similar CGCs are better at competing against R401 and providing protection. I think it would be helpful to more explicitly say that the genetic content of these strains likely drives this competition (at places like line 340-341 or 345 "resource usage profiles").

Figure 3B and lines 222-224 & 233-235. The statistical comparison seems to be within each group (axenic, AtSC6, LjSC6) and not across the whole figure. For example LjSC6 and ATSC6 are both "b" but look very different, while LjSC6 "b" looks very similar to AtSC6-A68+L59 "a". The authors should confirm this analysis is correct and add more description in the figure caption about if the comparison is across the whole figure or just groups. If the current comparisons are only within each group (axenic, AtSC6, LjSC6) then is the conclusion at line 222-224 still true?

Did the authors consider whether any of the Pseudomonas strains they test in the section from line 237-253 contain the pvdY and phlD genes deleted in the R401 mutant that are causal for the disease phenotype. For example are the most closely related phylogenetic Pseudomonas strains missing these genes or do they also have them?

Figure 5. What is the tree at the right of the figure? The figure caption should be expanded to include more description (line 296-298).

Minor comments:

The sentence at line 72-74 is confusing. I don't understand why "e.g. though biotic or abiotic stress alleviation" is being layered into this point about what strains a host accommodates.

At line 75 is "the origin" specifically the host species origin? Or some other type of origin.

It seems like the work of Cara Haney (cited as #45, and possibly other work from the Haney lab) could be relevant in the introduction related to Pseudomonas intra-genus competition in disease phenotypes.

I think it would help to say at line 165 what the medium for the liquid cultures is. I realize this is described in the methods, but because this experiment is such a central part to the work the medium this culturing is conducted in is an important detail for context (is it complex, minimal, root exudate based? Would be helpful for readers).

At line 201: "likely due to the presence of specific strains competing with R401"…. This is ok, but to me "competition" sounds like resource competition. What if strains are actively antagonizing R401 (by producing antibiotic compounds for example)? I think broadening the possibilities described here for explaining this result would be good to consider.

Figure 4. It is a hard to tell the light from dark green coloring in the circles. And line 264 what is meant by "either medium protection or slightly worse" more specific/precise language here would be helpful. Maybe should reference the pictures of the plants.

Line 522 "visual assessment of the culture density" is an interesting way to normalize the culture. This is probably ok if the input was analyzed (which appears to be the case… Figure S6 t0 bar). It may be useful to explicitly say that the input of these experiments was analyzed in the methods here.

Line 528 "samples were taken from all cultures" how?

Reviewer #3:

In this manuscript, the authors explore how potential microbial competition impacts protection against an opportunistic Pseudomonas strain. The work nicely builds upon previous understanding with this strain and uses important bacterial isolate collections to build a series of synthetic communities to uncover common catabolism patterns among protective Pseudomonas strains in their collection. There are a couple of additions/clarifications that would help connect aspects of the stories better:

1) The statement in lines 144-145 is a little confusing. It only highlights that AtSC5 and AtSC6 rescue detrimental phenotypes, but each SC has higher biomass and improved appearance compared to the MS controls in Fig 1A,B. Shouldn't all SCs be highlighted in this statement? Subsequent experiments focus on AtSC6 and LjSC6, so it seems strange to highlight AtSC5 in this statement, since it isn't the focus in the subsequent experiments.

2) Lines 106-108 generally talk about trends, but only LjSC5 has significantly lower R401wt colonization compared to the MS control. To identify members of LjSC5 that might directly inhibit R401 growth, shouldn't these members be examined for their microbe-microbe interactions with R401? Lines 129-131 mentions that Bacillus strain AtRoot131 inhibits R401, but what about the members of LjSC5 that haven't already been shown? There should be ~10 members that aren't redundant with LjSC6 or the Pseudomonas R154 strain highlighted at the end of the paper. These pairwise antagonistic assays might more clearly tie up this part of the manuscript more clearly before moving onto the subsequent seedling experiments.

3) Adding statistics to highlight the difference between WT and mut R401 in Fig 1D would make it easier to compare the results with Fig 2B. The authors move into very nice follow up experiments with Pseudomonas strains, but there are other interesting comparisons to make between the in vitro assays in Fig 1D and the seedling experiments in Fig 2B to highlight which interactions might be plant-dependent vs. plant-independent.

4) Are there statistically supported differences for Fig 2C? If there is not a significant difference between wt and mut R401, it's not really accurate to say the mutant is partially impaired.

5) Is there any biomass to support the findings for Fig 4? LjRoot54 and LjRoot 154 appear to be an intermediate phenotype between R401 and the others.

6) The findings in Fig 5 are really interesting and highlight that there is not much overlap in the metabolites present in the artificial root exudates used for the growth curves with these metabolites. Were there any follow up growth curves tried with these compounds to confirm the proposed metabolic competition that is suggested in the manuscript?

Reviewer #4:

Here, Amrhein et al. present a nice study investigating the protective ability of SynComs against an opportunistic bacterial pathogen. The experiments are well described, well designed, and the results are clearly and accurately interpreted in my opinion. The authors pursue multiple routes to understand how suppression of R401 by individual Pseuodmonas isolates might be occurring yet find no 'smoking gun', highlighting the complexity of in host microbe-microbe dynamics and disease outcomes. The authors do a nice job discussing this point.

My only major comment is that the manuscript includes text in the introduction and discussion regarding host-specific selection of the microbiome, with the underlying assumption that this is adaptive. Counter to this idea is the observation that the AtSC is not protective against R401's virulence towards Arabidopsis. The authors discuss that perhaps since R401 originated from Arabidopsis, it has evolved the ability to compete more effectively against AtSc members, rather than LjSC members, and thus become prominent and cause disease in the presence of AtSC. However, Fig. 4 very nicely demonstrates that Pseudomonas isolate collections from either At or Lj contain both fully protective and non-protective members. So my critique here, and more broadly to the entire plant microbiome field, is that whether or not a microbe is host selected or not is, more often than not, entirely unrelated to host-derived benefits.

Minor comments:

Line 23: perhaps mention here where R401 was isolated from.

Line 87: Pseudomonas intra-genus competition as a possible strategy to control Pseudomonas pathogen.

Fig.1: please indicate the base used in the presentation of CFUs in log units

Fig.2/S5: I'm curious to see if the effect of R401 WT on AtSC6 composition are similar in plant roots as in liquid culture (when added at T0). This could be presented as either a full PCoA with all of the samples together or possibly similar to Fig. 2B where the fold change of individual SC members in response to R401 are shown in both roots and liquid culture. I don't think this has to be a main text figure.

Line 192-197: This is a fascinating observation. Another point that might be worth making is the lack of spatial structure in a shaken liquid culture versus in a plant root where perhaps R401 is able to invade relative unoccupied or differentially occupied habitats.

Line 219: "present in LjSC6 instead of LjRoot59 or when no other strain was added …after removal of L59".

Line 225: Can the authors speculate on what might happen to the suppression of R401 if A68 and L59 co-occurred? Any idea of whether A68 may interfere with the suppressive capacity of L59 through competition?

Line 285: I'd like a bit more detail on the generation of the CGC profiles. For a given metabolite in Fig. 5, e.g. anthranilate, there are two strains with 1 CGC. Do they share the identical CGC? Is this considered in the Euclidean distance calculation?

---

## [Decision Letter · Decision Letter 2]

Dear Kathrin,

Thank you for your patience while we considered your revised manuscript "Pseudomonas intra-genus competition determines protective function of SynComs in Arabidopsis thaliana" for publication as a Short Reports at PLOS Biology. This revised version of your manuscript has been evaluated by the PLOS Biology editors, the Academic Editor and three of the original reviewers.

Based on the reviews, we are likely to accept this manuscript for publication, provided you satisfactorily address the remaining editorial points. Please also make sure to address the following data and other policy-related requests.

a) The study has been peer-reviewed as a Short Report, which has a limit of 4 main Figures. You currently have 5 main figures. Please reduce them by either combining them, or sending one to the supplementary material.

b) We routinely suggest changes to titles to ensure maximum accessibility for a broad, non-specialist readership, and to ensure they reflect the contents of the paper. In this case, we would suggest a minor edit to the title, as follows. Please ensure you change both the manuscript file and the online submission system, as they need to match for final acceptance:

"Pseudomonas intra-genus competition determines the protective function of synthetic bacterial communities for Arabidopsis thaliana"

Please supply the numerical values either in the a supplementary file or as a permanent DOI’d deposition for the following figures:

Figure 1BCD, 2ABC, 3B, 4AB, 5, S4, S5AB, S7AB, S8ABCD, S9, S10, S11B, S13ABC

d) Please cite the location of the data clearly in all relevant main and supplementary Figure legends, e.g. “The data underlying this Figure can be found in S1 Data” or “The data underlying this Figure can be found in https://doi.org/10.5281/zenodo.XXXXX”

e) Please provide the tree files for the phylogenetic trees in Figures 4C, S1AB

f) Please ensure that your Data Statement in the submission system accurately describes where your data can be found and is in final format, as it will be published as written there.

g) Per journal policy, if you have generated any custom code during the course of this investigation, please make it available without restrictions upon publication. Please ensure that the code is sufficiently well documented and reusable, and that your Data Statement in the Editorial Manager submission system accurately describes where your code can be found.

We expect to receive your revised manuscript within two weeks.

*Published Peer Review History*

*Press*

Sincerely,

Melissa

Melissa Vazquez Hernandez, Ph.D.

Associate Editor

PLOS Biology

REVIEWERS' COMMENTS:

Reviewer #1: The authors have addressed all of my concerns with the initial version of the manuscript, as well as (in my opinion) those raised by the other three reviewers.

Reviewer #2: The authors have thoroughly addressed my previous comments and those of the other reviewers. The revised manuscript includes the requested clarifications on SynCom design, statistical comparisons, and genomic analyses of protective Pseudomonas strains. I appreciate the additional data (e.g., Supplementary Figure 1, ANI analysis) and the clarifications made throughout the text and captions.

---

## [Editor Report · Decision Letter 3]

Dear Kathrin,

Thank you for the submission of your revised Short Reports "Pseudomonas intra-genus competition determines the protective function of synthetic bacterial communities in Arabidopsis thaliana" for publication in PLOS Biology. On behalf of my colleagues and the Academic Editor, Cara Helene Haney, I am pleased to say that we can in principle accept your manuscript for publication, provided you address any remaining formatting and reporting issues. These will be detailed in an email you should receive within 2-3 business days from our colleagues in the journal operations team; no action is required from you until then. Please note that we will not be able to formally accept your manuscript and schedule it for publication until you have completed any requested changes.

PRESS

Sincerely, 

Melissa

Melissa Vazquez Hernandez, Ph.D., Ph.D.

Associate Editor

PLOS Biology
